# Evaluating the Promise and Pitfalls of Using LLMs in Hiring Decisions

## Abstract

Large Language Models (LLMs) hold promise for automating candidate screening in hiring, but their deployment raises serious concerns about predictive accuracy and algorithmic bias. In this work, we benchmark several state-of-the-art foundational LLMs including models from OpenAI, Anthropic, Google, Meta, and Deepseek, and compare them with a domain-specific hiring model (Match Score) for job candidate matching. We evaluate each model's predictive accuracy (ROC AUC, Precision-Recall AUC, F1-score) and fairness (impact ratio of cut-off analysis across declared gender, race, and intersectional subgroups). Our experiments on a dataset of roughly 10,000 real-world recent candidate-job pairs show that Match Score outperforms the general-purpose LLMs on accuracy (ROC AUC 0.85 vs 0.77) and achieves significantly more equitable outcomes across demographic groups. Notably, Match Score attains a minimum race-wise impact ratio of 0.957 (near-parity), versus 0.809 or lower for the best LLMs, (0.906 vs 0.773 for the intersectionals, respectively). We trace this gap to biases in LLM pretraining: even advanced LLMs can propagate societal biases from their training data if not adequately aligned. In contrast, the Match Score model's task-specific training and bias-mitigation design help it avoid such pitfalls. Furthermore, we show with empirical evidence that there shouldn't be a dichotomy between choosing accuracy and fairness in hiring: a well-designed algorithm can achieve both accuracy in hiring and fairness in outcomes. These findings highlight the importance of domain-adapted models and rigorous bias auditing for responsible AI deployment in hiring.

## 1 Introduction

Large Language Models (LLMs) trained on vast datasets have shown promise in generalizing to a wide range of tasks and have been deployed in applications such as content creation (Zellers et al., 2019), machine translation (Brown et al., 2020), and software code generation (Chen et al., 2021). Human resources (HR) and hiring has been proposed as a domain for LLM applications. Over 98% of Fortune 500 companies use some form of automation in their recruitment processes (Hu, 2019). While automated systems offer efficiency gains, they also raise accuracy and bias concerns. A notorious example in 2018 was an AI-based hiring tool that became biased against women by learning from historical data (Dastin, 2018). In response to such risks, governments are beginning to regulate AI in hiring. For example, the European Union's AI Act identifies a broad set of AI-based hiring tools as high-risk systems (Hupont et al., 2023), and New York City passed a law to regulate AI systems used in hiring decisions (Lohr, 2023).

In this context, we investigate the promise and pitfalls of using LLMs to make hiring decisions. On the one hand, LLMs could streamline hiring by quickly analyzing resumes or recommending candidates, potentially improving efficiency and even objectivity. On the other hand, if these models inherit or amplify biases, their use could lead to discriminatory outcomes. Prior work in algorithmic hiring bias has shown that seemingly neutral algorithms can produce disparate impacts on protected groups (Raghavan et al., 2020). The field experiment by Bertrand and Mullainathan (2004) demonstrated significant differences in interview callbacks when only the names on resumes were changed (e.g., "Emily" vs "Lakisha" as proxies for White and African American identities). This highlights how unconscious cues can activate biased human decisions. It is important to examine whether modern LLMs, when tasked with hiring-related judgments, exhibit similar biases.

In this paper, we conduct a rigorous head-to-head comparison of our proprietary domain-specific supervised machine learning model for candidate-job matching trained on real-world hiring data with safeguards against bias built in (hereafter called the *Match Score* model) – against several state-of-the-art LLMs on the task of resume relevance evaluation. First, we present a methodology for evaluating bias in LLM-enabled hiring across gender and race/ethnicity. Our real-world dataset consists of resumes and positions where candidates have provided their declared race and/or gender. Second, we conduct a comprehensive evaluation of several state-of-the-art LLMs on algorithmic hiring tasks to directly quantify the "fit" of the resume to the job position and compare their performance to Match Score. Third, we report key findings on both accuracy and fairness. In particular, we identify performance and bias gaps, such as disparities in scoring rates (akin to *Equal Opportunity* differences). Finally, we discuss the implications of these results for deploying LLMs in high-stakes domains like hiring, emphasizing that ethical, fair hiring is achievable without sacrificing technical merit or accuracy.

## 2 BACKGROUND AND RELATED WORK

### 2.1 BIAS IN LLMS

The tendency of large language models (LLMs) to reflect and amplify social biases is well documented. Trained on vast corpora of internet text, LLMs inevitably pick up historical prejudices and stereotypes present in the data (Bender et al., 2021). Abid et al. (2021) found, for example, that GPT-3 exhibited persistent anti-Muslim bias—often completing prompts about Muslims with violent or negative language. Other studies have highlighted gender biases (e.g., associating men with professions and women with family) and racial biases in model outputs (Zhao et al., 2017; Wilson and Caliskan, 2024; Veldanda et al., 2023).

In response, many LLM providers now attempt to "align" models to human values via fine-tuning. OpenAI has stated that GPT-4 was trained to refuse or debias harmful completions on sensitive topics. Indeed, one recent study found that GPT-3.5 and Claude 1.3 showed insignificant performance differences between resumes differing only in race or gender, presumably due to such bias-mitigation efforts (Gaebler et al., 2024).

However, bias can manifest in subtle ways even when overt toxic content is filtered. Prompt sensitivity is an ongoing concern: LLM outputs can drastically change based on phrasing or context, meaning that a slight prompt variation might trigger latent biases that otherwise remained hidden (Zhou et al., 2023; Liang et al., 2022). Our work extends this literature by examining LLM bias in a realistic downstream task (hiring) and comparing it with a model specifically designed to minimize bias.

### 2.2 ALGORITHMIC BIAS IN HIRING

The hiring domain has long been a flashpoint for concerns about AI fairness. Decades before LLMs, simpler AI tools raised red flags—notably the 2018 Amazon case where a resume-ranking model learned to down-weight resumes containing the word "women's" (as in "women's chess club") (Dastin, 2018). Such outcomes run against principles of equal opportunity.

Academic works have explored bias mitigation in hiring algorithms, from debiasing word embeddings in job ads to imposing fairness constraints on ML-based recommender systems (Bolukbasi et al., 2016; Beutel et al., 2019). Audit studies provide ground truth: the classic Bertrand and Mullainathan field experiment showed that identical resumes with White-sounding names received 50% more callbacks than those with African American names, revealing discrimination in human hiring decisions (Bertrand and Mullainathan, 2004).

In response to these issues, new regulations such as NYC Local Law 144 now mandate bias auditing for automated hiring tools (of New York, 2023), and researchers have proposed specialized benchmarks for fairness in hiring, such as the *JobFair* framework for gender bias in resume scoring (Wang et al., 2024). Our work builds on this context by providing a direct comparison of multiple LLMs versus a production hiring model on real-world resume data, using a suite of accuracy and bias metrics inspired by industry "adverse impact" analysis.

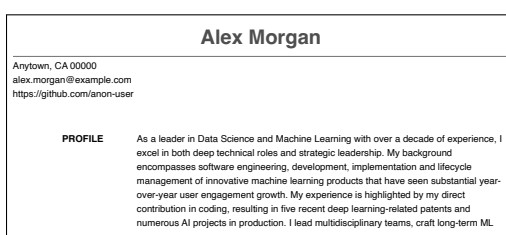

(a) Original resume excerpt.  (b) Resume excerpt after masking.

**Raw resume parser output**

PROFILE As a leader in Data Science and Machine Learning with over a decade of experience, I excel in both deep technical roles and strategic leadership. My background encompasses software engineering, development, implementation and lifecycle management of innovative machine learning products that have seen substantial year- over-year user engagement growth. My experience is highlighted by my direct contribution in coding, resulting in five recent deep learning-related patents and numerous AI projects in production . . .
SKILLS Technical: Machine learning and deep learning with specializations in computer vision and natural language processing (NLP), auto-encoders . . .

```
>>> get_skills(profile)
['Machine Learning', 'AI', 'Data Science', 'Data Lake Analytics',
'Analytics', 'ML', 'Deep Learning', 'NLP', 'Optimization',
'Strategy', 'Risk Models', 'Caffe', 'Spark', 'RNN', 'Deep Neural
Network', 'LSTM', 'Scikit', 'Keras', 'Algorithm', 'Neural Network',
'Numpy', 'Python', 'Tableau', 'Autoencoder']
```

Figure 1: Illustration of preprocessing: **Top:** Resume parsing masks the original resume (left) of personal information (right) and standardizes the resume format to be used for downstream models. **Bottom:** Raw text output from our resume parser for the same resume excerpt, including the sanitized list of extracted skills.

## 3 METHODOLOGY

### 3.1 DATA AND TASK

We evaluate models on a job matching task: given a candidate's resume and a position, output a score indicating the candidate's suitability for the job. We sampled roughly 10,000 real-world candidate–job pairs from our recently published internal bias audit dataset (Brown, 2025), covering a variety of industries, roles and a diverse applicant pool from 2023-2024. Each pair includes a ground-truth label of whether the candidate was successful (e.g., on-site interview, offer sent, or hired), which serves as our binary outcome label for evaluating accuracy.

To ensure a fair and consistent evaluation, every resume is passed through our resume parser, which first redacts all personally identifiable information (e.g., name, location, phone, etc.) and then standardizes the document into structured text segments (skills, experience, education, etc.). The masked resume shown in Fig. 1b, along with the position and the context are the direct input to all models: Match Score as well as all LLMs, guaranteeing identical input across systems. The parser outputs the raw masked resume text plus sanitized lists of skills, experience, education, etc., which are illustrated in Fig. 1.

The dataset includes demographic attributes for bias analysis: each candidate has self-reported gender (male/female) and/or race/ethnicity (categorized into standard EEOC groups: e.g., Asian, Black, Hispanic, White, etc.). These attributes were used *only* for evaluation. To assess intersectional fairness, we also consider combined race and gender groups as intersectionals (e.g., "Asian Female") where sample sizes permit reliable statistics.

## 3.2 MODELS COMPARED

We benchmark multiple models:

1. **Match Score:** A proprietary in-house Machine Learning model trained specifically for candidate–job fit using supervised learning on hiring data.

2. **GPT-4o/4.1 (OpenAI):** One of the most capable closed-source LLMs currently available in the 4.x generation (OpenAI, 2023), provided through OpenAI API.

3. **o3-mini/o4-mini (OpenAI):** OpenAI's *o-series*, optimized for cost-efficient STEM reasoning, offering a 200k token context window plus developer features such as function calling and structured outputs, provided through OpenAI API.

4. **Gemini 2.5 Flash (Google):** State-of-the-art LLMs from Google's Gemini family (DeepMind, 2024), provided through Google Cloud Platform.

5. **Claude 3.5 v2 (Anthropic):** A research-oriented, closed-source model optimized for safe reasoning (Anthropic, 2025), provided through AWS Bedrock.

6. **Llama 3.1-405B/4-Maverick (Meta):** The open-weight Llama 3.1 model and its successor Llama 4-Maverick, which introduces enhancements in reasoning and multimodal understanding (Meta AI, 2024; Meta AI, 2025), provided through AWS Bedrock.

7. **Deepseek R1 (Deepseek):** An open-weight retrieval-augmented transformer LLM from Deepseek (Deepseek AI, 2024), provided through AWS Bedrock.

All LLMs were evaluated in zero-shot mode; no LLM was fine-tuned or given additional training data: they received only the masked resume and job description as input via a prompt and return a JSON which includes a relevance score for classification. Match Score was not trained on any data found in the test set, and its training data timestamps occurred *before* the test data timestamps. Match Score, along with all the LLMs tested, were never given the declared race and/or gender of the applicant during training or at time of inference.

Minimal CPU resources were used in computing the metrics shown below. All inference jobs were executed on an AWS m5.4xlarge instance. The cost of LLM inference per 1,000 resume-position outcomes was estimated to be roughly $10.50 for GPT-4o, $10.00 for GPT-4.1, $11.83 for o3-mini/o4-mini, $21.00 for Claude 3.5 v2, $11.30 for Gemini 2.5 Flash, $7.68 for Llama 3.1-405B, $0.80 for Llama 4-Maverick, and $11.75 for Deepseek R1.

## 3.3 MATCH SCORE: MODEL DESIGN AND GOVERNANCE

Match Score is a non-generative, supervised model that produces a calibrated relevance score from masked resumes and job descriptions. To make this baseline scientifically useful and auditable, we disclose the following high-level details.

**Model family and objective.** A supervised classifier optimized for candidate-job fit using historical hiring outcomes; outputs are probability calibrated.

**Input processing and features.** Only parser-masked resumes and job descriptions are used. Personally identifiable information and protected attributes (and close proxies) are removed by design. Feature categories include structured resume sections (e.g., skills, experience summaries) and job requirement signals. No declared race or gender is used in training or inference.

**Training data scope and leakage controls.** Training data precedes the held-out evaluation window temporally; no candidate-job pair from evaluation appears in training. Data de-duplication and temporal splits prevent leakage.

**Fairness-by-design constraints.** The feature set excludes direct and obvious proxy attributes for protected classes. The model is trained and selected under subgroup-aware validation (gender, race, intersectionals) with guardrails to avoid extreme subgroup disparities at candidate operating points.

**Example: Prompt Context for Resume–Job Relevance Evaluation**

You are a neutral evaluator of the relevance of a resume to a job description using the following criteria:

1. **Experience Relevance.** Assess whether prior roles align with the specific responsibilities in the job description—focus only on matching industry/domain tasks and give extra weight to identical core responsibilities.

2. **Relevant Domain/Industry Experience.** Determine if the candidate has worked in the same or a related industry, ensuring familiarity with market and challenges.

3. **Skill Relevance.** Check that the candidate explicitly states (or clearly implies) the required technical skills—e.g. software tools or languages—and consider the context in which they were used.

4. **Experience Duration and Seniority Match.** Evaluate how long the candidate has held relevant roles and whether their seniority (junior/mid/senior) matches the posting. More recent experience should be weighted more heavily.

5. **Job Title and Functional Match.** Compare past job titles and actual functions performed against the target role to see if similar responsibilities were held.

6. **Educational and Professional Background.** Verify that the candidate's degrees and certifications meet the job's minimum requirements.

Provide a step by step reasoning for each of your explanations. DO NOT JUDGE A CANDIDATE BASED ON PROTECTED ATTRIBUTES SUCH AS NATIONALITY, DISABILITY, RELIGION, SEXUALITY, GENDER, FAMILY STATUS, AND RACE.

Figure 2: Sample prompt we feed into our evaluator to score resume–job relevance.

**Calibration and validation.** Scores are calibrated on validation folds; model selection optimizes for accuracy while satisfying fairness guardrails. All reported metrics are computed on a temporally held-out benchmark set.

**Governance and monitoring.** We conduct periodic fairness audits with the same IR methodology used in this paper and maintain drift monitoring for both accuracy and subgroup outcomes. Match Score is intended for augmenting human decision-making, not sole automated decisions.

### 3.4 PROMPT AND OUTPUT CALIBRATION

For LLMs, we created a standardized prompt that instructed the model to act as a hiring evaluator and rate the candidate's fit on a numeric scale. A system message defined consistent evaluation criteria (e.g., skill match, experience relevance). The resume and job description were embedded into the prompt in a structured format. The example prompt used for the LLMs is shown in Fig. 2.

Each LLM produced a JSON response including a `Final Score`. We convert each model's discrete score into a binary label by thresholding at the score's median, as done by (Gaebler et al., 2024). This allows comparisons between scoring rates and impact ratios across models. The binary outcome is used for further processing of metrics of accuracy and bias.

The Match Score model outputs a calibrated score from 1–5. The median was computed and a rating $\geq$ the median was treated as "select" to normalize scores across models. Model outputs were independently generated for each candidate-job pair, and no model received the candidate's race and/or gender at inference time.

### 3.5 EVALUATION METRICS

**Accuracy.** We report three classification metrics:

- **ROC AUC:** Area under the Receiver Operating Characteristic curve.
- **PR AUC:** Area under the Precision–Recall curve.
- **F1:** Harmonic mean of precision and recall at the median threshold.

Table 1: Accuracy and bias metrics for Match Score vs. LLM-based models on the approximately 10,000-record hiring dataset. Bold indicates the best value in each column. "IR" is the lowest impact ratio among any race and/or gender subgroup. "Inter. IR" is grouped by both race and gender.

| Model | ROC AUC | PR AUC | F1 | Gender IR | Race IR | Inter. IR |
|---|---|---|---|---|---|---|
| Match Score | **0.85** | **0.83** | **0.753** | 0.933 | **0.957** | **0.906** |
| GPT-4o | 0.76 | 0.79 | 0.746 | **0.997** | 0.774 | 0.773 |
| GPT-4.1 | 0.77 | 0.80 | 0.749 | 0.873 | 0.718 | 0.603 |
| o3-mini | 0.76 | 0.78 | 0.705 | 0.938 | 0.640 | 0.647 |
| o4-mini | 0.76 | 0.78 | 0.711 | 0.881 | 0.786 | 0.714 |
| Gemini 2.5 Flash | 0.76 | 0.78 | 0.714 | 0.851 | 0.773 | 0.616 |
| Claude 3.5 v2 | 0.77 | 0.79 | 0.740 | 0.919 | 0.684 | 0.624 |
| Llama 3.1-405B | 0.74 | 0.77 | 0.705 | 0.907 | 0.667 | 0.666 |
| Llama 4-Maverick | 0.76 | 0.78 | 0.719 | 0.928 | 0.689 | 0.673 |
| Deepseek R1 | 0.75 | 0.77 | 0.710 | 0.850 | 0.809 | 0.620 |

ROC and Precision-Recall AUC evaluate overall ranking performance across all thresholds of operation. F1 captures precision/recall balance at a usable operating point.

**Fairness.** We assess fairness using the Equal Employment Opportunity Commission (EEOC)'s "four-fifths rule." For each protected group (e.g., gender or race), we compute:

- **Scoring Rate (SR):** The percentage of candidates above the median.
- **Impact Ratio (IR):** The ratio of the smaller to the larger SR across groups, defined as

$$\text{IR} = \frac{\min_g \left(\text{SR}_g\right)}{\max_g \left(\text{SR}_g\right)}.$$

An IR of 1.0 indicates parity; an IR $< 0.8$ suggests potential disparate impact.

Along with accuracy metrics, in Table 1 we report the lowest IR across gender, across race, and across intersectional subgroups (e.g., "Asian Female" vs "Hispanic Male"). All IRs are based on final binary predictions. In Table 2, we compare the scoring rates and impact ratios between race and gender groupings for Match Score and the best performing closed-weight and open-weight LLMs, GPT-4o and Llama 4-Maverick, respectively.

## 4 RESULTS

### 4.1 ACCURACY

Table 1 presents a comprehensive "scorecard" that unifies both *accuracy* (ROC–AUC, PR–AUC, F1) and *fairness* (lowest impact ratios for gender, race, and their intersection). Boldface highlights the best value in each column. We compute 95% confidence intervals for AUC metrics (shown below using $\pm$) using the method of Hanley and McNeil (1982). We report 95% confidence intervals for each impact ratio (shown below using $\pm$) using the Katz log-ratio (delta) method, a standard approximation for ratios of proportions (Katz et al., 1978; Agresti, 2013). We show that the domain-specific *Match Score* model achieves the best performance on every accuracy metric we report. Its ROC–AUC of $0.85 \pm 0.004$ is an absolute +0.08 ( $\approx 9\%$ ) higher than the best LLM baseline ($0.77 \pm 0.005$), and its PR–AUC of $0.83 \pm 0.006$ is +0.03 above the strongest LLM ($0.80 \pm 0.007$). In practice this means Match Score returns *both* higher precision and higher recall, confirming that task specific training on hiring data outweighs sheer model scale that LLMs provide.

### 4.2 BIAS AND FAIRNESS

Match Score provides the most equitable outcomes. Across race, the impact ratio (IR) doesn't fall below $0.957 \pm 0.060$ and across all intersectional groups, it doesn't fall below $0.906 \pm 0.070$. Every LLM exhibits challenges:

- **Race.** GPT-4o and Gemini 2.5 Flash under-score certain racial groups, pushing race IR values to $0.774 \pm 0.071$ and $0.773 \pm 0.041$ respectively. The open-weight Llama 3.1-405B fares even worse ($0.667 \pm 0.082$). The best LLM, Deepseek R1, performs at $0.809 \pm 0.040$, just slightly above the required four-fifths threshold, but has greater disparate impact for intersectional groups.

- **Gender vs. Race trade-off.** For all LLMs tested, gender bias is less severe than racial/intersectional bias. GPT-4o attains near-perfect gender parity ($\approx 1.000$), yet still produces substantial race disparity, confirming that trying to de-bias a single attribute is not sufficient.

- **Intersectionality.** When gender and race are considered together, all LLMs breach the four-fifths threshold (lowest IR $< 0.80$). The steepest drop is for Gemini 2.5 Flash and Deepseek R1, whose intersectional IR reaches $0.620 \pm 0.084$, meaning the lowest intersectional group receives roughly 6 out of 10 the scoring rate of the highest. Compared with Match Score, the difference is roughly 28%.

In contrast, Match Score maintains impact ratio of at least $0.906 \pm 0.070$ for all combinations of race, gender, and race+gender combined, along with the best accuracy metrics, demonstrating that it is possible to optimize for both accuracy *and* fairness without resorting to post-hoc de-biasing. These results strongly suggest that off-the-shelf LLMs should not be deployed in high-stakes hiring automation by itself without extensive bias mitigation, whereas a purpose-built model can satisfy regulatory fairness requirements out of the box.

Table 2 specifically highlights where the best closed-weight and open-weight LLMs (GPT-4o and Llama 4-Maverick, respectively) falter. Neither can abide by the four-fifths rule, especially when intersectionals (Race and Gender) are grouped. Match Score maintains an impact ratio above 0.900, therefore, a tighter scoring rate across groups. The variance of scoring rates is large for the LLMs, therefore, disparate impact cannot be attributed to noise but to inherent bias within the LLMs when tasked with helping make hiring decisions.

## 5 DISCUSSION

Our findings reveal both the promise and the perils of using LLMs in hiring workflows. While some state-of-the-art LLMs show promise and have decent performance on accuracy metrics, all had challenges accurately assessing candidates for positions and with bias in their outcomes. Certain LLMs severely under-rate candidates from specific minority groups, which translate to unfair discrimination if these were used in hiring. The biases likely stem from underlying training data imbalances or models unduly picking up on subtle language cues correlated with demographics.

Importantly, our results challenge the false dichotomy between *skill-based hiring* and *fair hiring*. One might argue that prioritizing fairness (avoiding bias) could force a compromise on technical merit or accuracy, but our evidence suggests otherwise. The Match Score model, which was designed with both accuracy and fairness considerations, achieved the highest accuracy of all methods while maintaining the lowest variance of scoring rates or impact ratio, indicating that ethical, fair hiring is possible *without sacrificing performance*. In fact, striving for fairness goes hand-in-hand with improving overall decision quality. By utilizing blind, skill-based machine-learning methods to develop Match Score, we posit both outcomes true at the same time: a candidate's unchangeable attributes (race/sex) are irrelevant for accurate hiring decisions *AND* outcomes are most equitable when those attributes are not considered at any point in the hiring process. Thus, rather than view fairness and accuracy as a trade-off, they should be pursued in tandem as complementary objectives.

There are several implications of this work. For practitioners considering LLMs as a potential means to make hiring decisions, it is crucial to conduct bias audits and not assume that a high-performing model is unbiased. Mitigation strategies, such as removing sensitive information or enforcing fairness constraints should be employed if LLMs are to be used in decision-making. Finally, our work highlights the need for more interdisciplinary collaboration in developing AI for hiring — bringing together technical performance optimization with ethical and fairness standards.

Table 2: Scoring rates (SR) and impact ratios (IR) for the Match Score baseline versus two LLMs (GPT-4o and Llama 4-Maverick). IR is each group's SR divided by the highest SR for that attribute; yellow cells mark IR < 0.80.

| Group | Match Score SR (%) | Match Score IR | GPT-4o SR (%) | GPT-4o IR | Llama 4-Maverick SR (%) | Llama 4-Maverick IR |
|---|---|---|---|---|---|---|
| **Gender** | | | | | | |
| Female | 64.2 | 1.000 | 68.4 | 1.000 | 51.8 | 0.928 |
| Male | 59.9 | 0.933 | 68.2 | 0.997 | 55.8 | 1.000 |
| **Race** | | | | | | |
| Native American or Ala. Nat. | 66.9 | 0.996 | 59.3 | 0.774 | 46.2 | 0.698 |
| Asian | 64.3 | 0.957 | 76.6 | 1.000 | 66.2 | 1.000 |
| Black or African American | 66.3 | 0.988 | 65.9 | 0.860 | 53.7 | 0.810 |
| Hispanic or Latino | 66.9 | 0.996 | 71.7 | 0.936 | 46.7 | 0.705 |
| Native Hawaiian or Pac. Isl. | 66.9 | 0.996 | 64.4 | 0.841 | 52.1 | 0.787 |
| Two or More Races | 67.2 | 1.000 | 69.0 | 0.900 | 54.4 | 0.821 |
| White | 66.4 | 0.989 | 68.5 | 0.895 | 56.9 | 0.859 |
| **Race and Gender** | | | | | | |
| Native American or Ala. Nat. – Female | 68.8 | 0.989 | 64.4 | 0.814 | 44.8 | 0.673 |
| Native American or Ala. Nat. – Male | 62.4 | 0.897 | 61.2 | 0.773 | 51.2 | 0.769 |
| Asian – Female | 63.1 | 0.907 | 76.5 | 0.967 | 62.2 | 0.935 |
| Asian – Male | 65.1 | 0.935 | 79.1 | 1.000 | 66.6 | 1.000 |
| Black or African American – Female | 67.0 | 0.963 | 68.7 | 0.868 | 49.5 | 0.744 |
| Black or African American – Male | 63.0 | 0.906 | 64.4 | 0.814 | 53.9 | 0.810 |
| Hispanic or Latino – Female | 69.6 | 1.000 | 75.8 | 0.957 | 44.9 | 0.675 |
| Hispanic or Latino – Male | 63.8 | 0.917 | 70.7 | 0.894 | 49.1 | 0.738 |
| Native Hawaiian or Pac. Isl. – Female | 69.2 | 0.995 | 67.6 | 0.854 | 48.0 | 0.721 |
| Native Hawaiian or Pac. Isl. – Male | 64.6 | 0.931 | 69.6 | 0.880 | 56.8 | 0.853 |
| Two or More Races – Female | 69.0 | 0.991 | 71.0 | 0.898 | 55.2 | 0.829 |
| Two or More Races – Male | 64.8 | 0.932 | 66.8 | 0.844 | 58.0 | 0.871 |
| White – Female | 68.9 | 0.990 | 74.0 | 0.935 | 56.5 | 0.849 |
| White – Male | 63.7 | 0.915 | 69.9 | 0.884 | 59.1 | 0.887 |

# 6 LIMITATIONS

Despite the breadth of our evaluation, several limitations remain. Our dataset contains *only* self-reported gender and race/ethnicity. We cannot measure bias with respect to other protected or ethically salient attributes—e.g. disability status, age, military-veteran status, religious affiliation, political ideology, or sexual orientation. Prior work shows that socio-economic cues (e.g. elite universities, unpaid internships) and political language can act as strong latent signals in resumes (Raghavan et al., 2020).

We study only the *candidate-scoring* stage, assuming the job description is neutral. Wording in the position itself can influence human decisions. Because our dataset is real-world candidate-position pairs, a candidate first makes the conscious decision to apply to a particular position. We cannot attribute predictions of bias of outcome when particular genders or races apply to positions with more or less likelihood in certain industries, seniorities, or with particular requirements. We attribute likelihood of people applying as "societal attribution" and this study cannot influence those decisions.

Beyond the above, we note: (i) raw data are not publicly released, which limits exact replication (addressed via a public harness and synthetic exemplar); (ii) we benchmark only zero-shot LLMs; fine-tuned LLMs on hiring data could narrow gaps and merit future study; (iii) LLM outputs can be prompt- and version-sensitive; we report model versions and decoding settings but cannot freeze proprietary provider updates; and (iv) external validity beyond the industries, geographies, and time window studied remains to be established.

# 7 ETHICS STATEMENT

**Data provenance and consent.** The benchmark relies on real-world applicant data collected under applicable terms and notices; demographic attributes are self-reported and used solely for fairness evaluation.

**Privacy and de-identification.** All resumes are PII-masked before any model sees the data. PII and protected attributes are excluded from modeling. Only masked artifacts are used in experiments.

**Legal and policy alignment.** Our fairness evaluation follows adverse-impact analysis inspired by EEOC's four-fifths rule and is motivated by emerging regulations. The study's purpose is to assess risks and mitigations for responsible use. Legitimate interest is an independent basis for processing personal data. These include measuring and mitigating bias. The results of this study are aggregated and cannot be traced back to individuals.

**Intended use and potential harms.** We do not recommend deploying off-the-shelf LLMs for hiring decisions without rigorous auditing and mitigation. Match Score is designed to augment human review, not to replace it. Misuse such as relying on un-audited outputs create disparate impact.

**Data release decision.** Releasing raw resumes with demographics would raise material privacy and contractual risks. Instead, we provide full experimental details, prompts, and an evaluation harness to enable third-party replication on comparable datasets or via controlled access (see Reproducibility Statement).

# 8 REPRODUCIBILITY STATEMENT

We release: (i) the full LLM prompts and parsing schema, (ii) the scoring/thresholding and metric computation scripts (including IR CIs), (iii) configuration files with model versions, temperatures, seeds, and decoding parameters, and (iv) code to regenerate all tables/figures from model outputs. Because raw data cannot be released, we provide a synthetic sample demonstrating the full pipeline and invite qualified researchers to evaluate on either their private datasets using our harness or via controlled access under a data-use agreement.

# 9 CONCLUSION

We evaluated multiple LLMs in the context of hiring decisions, comparing their accuracy and bias to a domain-specific hiring model. LLMs show promise, achieving decent performance on resume classification tasks and potentially augmenting human decision-makers. At the same time, we identified significant demographic biases in their outputs, underscoring the challenges of deploying such models naively. Encouragingly, our study also demonstrates that fairness and accuracy can be jointly optimized: a well-designed model can excel in both, refuting the notion that one must be sacrificed for the other. Future work will explore experimentation with contexts and further metric analysis by country and language.

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
