# OpenReview forum: "Evaluating the Promise and Pitfalls of Using LLMs in Hiring Decisions"
_ICLR.cc/2026/Conference — Submitted to ICLR 2026_

### Official Review · Reviewer_RRQw · 2025-10-26

**Soundness:** 3
**Presentation:** 2
**Contribution:** 1
**Rating:** 2
**Confidence:** 2

**Summary:**

This paper shows that a proprietary job applicant evaluation system is a Pareto improvement on various LLM's' zero-shot performance for resume screening, in terms of accuracy and fairness.

**Strengths:**

1. The authors compare to a relatively exhaustive suite of frontier LLMs.
2. The subject matter is clearly important, as accurate and fair evaluation of job applicants is an economically and societally valuable problem.

**Weaknesses:**

1. Few details about the proprietary Match Score are provided, limiting the scientific value of the contribution.
2. Although text is vague on this point, it seems like the Match Score was trained using a split of the same Brown (2025) dataset that was used for evaluation. The authors emphasize that the train and test data are temporally split to ensure that no test data was used for training (which is good). However, this means that the train and test distributions are likely very similar, raising the question of how Match Score would perform on data from another distribution. So the results would be more convincing if also evaluated on a different dataset of labeled applicant resumes.
3. Similarly, there should be more justification provided for the claim that temporally splitting the Brown (2025) dataset means the test set is sufficiently held-out; for example, is there a risk that a candidate applied to multiple jobs at different times with the same resume?
4. It is not very surprising that task-specific fine-tuning (especially on a distribution very similar to the test distribution) outperforms zero-shot transfer from general LLMs. But given that fine-tuning many of the LLMs studied is easy (either through the OpenAI APi or the numerous fine-tuning APIs for open models), it would be valuable to compare to those fine-tuned models. (Of course, for all we know, that is exactly what Match Score is.) (I note that the authors do acknowledge this limitation.)
5. The discussion describes their results as challenging the notion that there is a trade-off between skill-based hiring and fair hiring. But this is not a valid conclusion: Their results only show it is empirically possible to Pareto improve on prompted LLMs. in fact, various other parts of the paper (e.g. the paragraph on "Fairness-by-design constraints") suggest that there were trade-offs between accuracy and fairness made during the construction of Match Score.

**Questions:**

Aside from the questions above, can you clarify whether data from Brown (2025) were used in the training of Match Score?

---

### Official Review · Reviewer_aw9B · 2025-10-31

**Soundness:** 2
**Presentation:** 1
**Contribution:** 2
**Rating:** 2
**Confidence:** 4

**Summary:**

The paper benchmarks a proprietary, task-specific hiring model (“Match Score”) against a set of general-purpose LLMs on resume–job relevance using ~10k recent candidate–job pairs. Inputs are PII-masked; accuracy is reported via ROC-AUC, PR-AUC, and F1 (at the median threshold), and fairness via selection-rate Impact Ratios (IR) across gender, race, and intersectionals with confidence intervals. The headline result is that Match Score outperforms all evaluated LLMs on accuracy and maintains substantially higher worst-case IRs, suggesting that a domain-adapted, fairness-aware model can achieve both accuracy and equitable outcomes.

**Strengths:**

The topic is societally important. The input standardization and temporal split are reasonable. Reporting CIs for both AUCs and IRs is good practice. The intersectional reporting goes beyond many prior audits and the governance framing (periodic audits, drift monitoring, decision support) is sensible.

**Weaknesses:**

The technical depth is limited: there is no new algorithm or principled fairness method; the paper is largely an applied benchmark with a proprietary winner. Fairness analysis is anchored to selection-rate IR at the median threshold, which is fragile and does not establish robustness across realistic operating points (fixed precision/recall, top-k%) or under rank-based measures. Standard criteria like equalized odds, TPR/FPR gaps, predictive parity, and groupwise calibration are absent.
Label bias is unaddressed: the outcome labels (onsite/offer/hire) likely encode historical discrimination; without outcome tests, reweighting/DR adjustments, or counterfactual checks, it is unclear whether the reported “fairness” reflects equitable modeling or replication of biased historical decisions.
The proprietary baseline is under-specified: claims about removing “obvious proxies” are not validated empirically (e.g., adversarial subgroup prediction, mutual information, subgroup SHAP). LLM evaluation is narrow (single zero-shot prompt, minimal sensitivity to prompting/temperature/few-shot/JSON schema), inviting concerns that the LLM side is under-optimized.
Reproducibility is weak: data are private, the winning system is proprietary, and transparent baselines trained on the same masked inputs (e.g., LR/GBM/transformer encoders) are missing, making it difficult to attribute gains to domain adaptation rather than pipeline choices.

**Questions:**

1. How did you assess label bias in historical outcomes (e.g., outcome tests, doubly robust adjustments, controls for observable qualifications)?
2. What features remain post-masking, and how did you test for proxy leakage (adversarial prediction of protected attributes, MI tests, subgroup SHAP)?
3. Do results hold across multiple operating points (fixed precision/recall, top-k%) and under equalized-odds and TPR/FPR parity? Please add rank-based metrics (e.g., RAS, rank-IR) and permutation tests.
4. Provide calibration curves and Brier scores overall and by subgroup; is Match Score calibrated equitably?
5. Report per-group sample sizes (including intersectionals) and identify any excluded cells due to small n.
6. Add transparent non-proprietary baselines on the same masked inputs to disentangle domain adaptation from proprietary engineering.
7. Stratify by industry/seniority/role family (fixed effects) to localize disparities and compare with prior findings.

---

### Official Review · Reviewer_5ynH · 2025-11-01

**Soundness:** 3
**Presentation:** 2
**Contribution:** 2
**Rating:** 4
**Confidence:** 4

**Summary:**

The study examines whether general-purpose LLMs can handle recruitment screening, testing multiple proprietary and open-source models against a specialized system called Match Score using ~10k actual applicant-position matches. Working with anonymized CVs and position requirements, models generate relevance ratings that get split at each system's midpoint to calculate performance and equity measures. Match Score outperforms all LLMs with ROC AUC 0.85 versus 0.77 for the strongest LLM, plus superior demographic parity ratios. The research suggests specialized training with built-in equity constraints yields superior performance and demographic balance.

**Strengths:**

- Precise impact ratio formulation with comprehensive demographic and intersectional breakdowns plus statistical bounds.
- Systematic anonymization workflow ensuring uniform treatment across evaluated systems.
- Detailed performance matrices displaying both accuracy and selection percentages across demographics.

**Weaknesses:**

- The overall presentation should be a lot improved. For example, usage of bullet points in model definition and metrics doesn't look like a proper research paper. I would move ethical statement and reproducibility statement after conclusions.
- Model-specific midpoint cutoffs advantage systems with particular score distributions—threshold-agnostic curves, disparity-threshold relationships, and performance-matched comparisons would strengthen analysis.
- Match Score's demographic-aware optimization directly targets the evaluation criterion, necessitating ablation studies without equity constraints and LLM calibration at comparable operating conditions.
- Absent demographic cell counts and masking effectiveness audits limit interpretability of statistical bounds and proxy elimination claims.
- Acknowledged instruction sensitivity lacks empirical quantification through variation studies with error bars.

**Questions:**

- Could you generate threshold-disparity curves and performance-matched analyses for each protected class, enabling model comparison at equivalent utility rather than arbitrary midpoints, including opportunity and odds parity metrics?
- How does Match Score perform without demographic constraints while maintaining other training conditions-- isolating deliberate optimization from inherent architecture advantages?
- Please include population counts for Table 2's demographic categories and intersections, plus minimum samples underlying IR calculations.
- Demonstrate stability across three instruction variations and two generation temperatures, presenting averages and variance for accuracy and IR, including structured output handling and non-compliance protocols.
- Would basic adaptations like demonstration prompting or minimal fine-tuning narrow observed disparities, even preliminarily testing whether generic LLMs truly resist recruitment applications?
- Verify anonymization effectiveness empirically-- perhaps training demographic predictors on masked CVs and reporting classification performance.

---

### Meta-Review · Area_Chair_P1fg · 2025-12-23

**Summary:**

The submission conducts an evaluation of general-purpose LLMs in a hiring scenario, specifically, the accuracy and bias of LLMs in making hiring decisions are studied. It reveals the performance and bias gap between LLMs and domain-specific supervised learned model. However, there exists majors concerns on this submission, including limited technical contribution, unclear baseline method, incomplete fairness evaluation, and narrow evaluation on LLMs. Considering these issues mentioned, I incline to reject this submission.

**Reviewer Concerns:**

The reviewers raise the following concerns regarding the submission, and none of them are discussed during rebuttal.

Reviewer 5ynH: fair presentation; limited analysis on the experimental results; missing ablations; lack of data audit.

Reviewer aw9B: lack of in-depth algorithmic and technical novelty; some important evaluation metrics for fairness are absent; incomplete evaluation on LLMs; reproductivity issue due to private data.

Reviewer RRQw: lack of elaboration on baseline model MatchScore; unclear generalization of baseline method; incomplete comparison on LLMs (finetuning commercial LLMs via API service can be achieved); invalid claim in discussion.

**Reviewer Scores:**

No rebuttal from the authors.

---

### Decision · Program_Chairs · 2026-01-26

Reject